A streptomycin resistance marker in H. parasuis based on site-directed mutations in rpsL gene to perform unmarked in-frame mutations and to verify natural transformation

Dai Ke 1
Wen Xintian xintian3211@126.com 1
Chang Yung-Fu 2
Cao Sanjie 1
Zhao Qin 1
Huang Xiaobo 1
Wu Rui 1
Huang Yong 1
Yan Qigui 1
Han Xinfeng 1
Ma Xiaoping 1
Wen Yiping yueliang5189@163.com 1
1 College of Veterinary Medicine, Sichuan Agricultural University , Chengdu , China
2 Department of Population Medicine and Diagnostic Sciences, College of Veterinary Medicine, Cornell University , Ithaca, NY , United States of America
Silva Pedro
Electronic publication date: 2018 Jan 11
Publication date: 2018
Volume: 6
Electronic Location ID: e4253
Received 2017 Oct 17; Accepted 2017 Dec 19
Copyright: ©2018 Dai et al.
Copyright year: 2018
Copyright holder: Dai et al.
License: This is an open access article distributed under the terms of the Creative Commons Attribution License, which permits unrestricted use, distribution, reproduction and adaptation in any medium and for any purpose provided that it is properly attributed. For attribution, the original author(s), title, publication source (PeerJ) and either DOI or URL of the article must be cited.
License URL: https://creativecommons.org/licenses/by/4.0/

Keywords: rpsL, Point mutation, EMS, Haemophilus parasuis, Streptomycin resistance, Natural transformation

Funding: Ministry of Agriculture of the People’s Republic of China (MOA) 201303034 This work was supported by the Ministry of Agriculture of the People’s Republic of China (MOA) (201303034). The funders had no role in study design, data collection and analysis, decision to publish, or preparation of the manuscript.

==============================
Haemophilus parasuis is a member of the family Pasteurellaceae and a major causative agent of Glässer’s disease. This bacterium is normally a benign swine commensal but may become a deadly pathogen upon penetration into multiple tissues, contributing to severe lesions in swine. We have established a successive natural transformation-based markerless mutation system in this species. However, the two-step mutation system requires screening of natural competent cells, and cannot delete genes which regulate natural competence per se. In this study, we successfully obtained streptomycin-resistant derivatives from H. parasuis wild type strain SC1401 by using ethyl methane sulfonate (EMS, CH3SO2OC2H5). Upon sequencing and site-directed mutations, we uncovered that the EMS-induced point mutation in rpsL at codon 43rd (AAA → AGA; K43R) or at 88th (AAA → AGA; K88R) confers a much higher streptomycin resistance than clinical isolates. We have applied the streptomycin resistance marker as a positive selection marker to perform homologous recombination through conjugation and successfully generated a double unmarked in-frame targeted mutant 1401D88△tfox△arcA. Combined with a natural transformation-based knockout system and this genetic technique, multiple deletion mutants or attenuated strains of H. parasuis can be easily constructed. Moreover, the mutant genetic marker rpsL and streptomycin resistant phenotypes can serve as an effective tool to select naturally competent strains, and to verify natural transformation quantitatively.

Introduction

Haemophilus parasuis (H. parasuis, HPS) is a Gram-negative, non-spore-forming, pleomorphic bacterium which is the etiological agent of Glässer’s disease. NAD (V factor), but not X factor, is necessary for the in vitro growth of H. parasuis in that the gene nadV, which encodes the nicotinamide phosphoribosyltransferase responsible for NAD salvage, is absent from the whole genome (Xu et al., 2011). The resulting illness is characterized by polyarthritis, fibrinous polyserositis and meningitis in pigs, producing significant mortality and morbidity in pig farms and leading to serious economic losses in the pork industry throughout the world (Oliveira & Pijoan, 2004). At least fifteen serotypes of varying virulence levels have been identified in this species (Kielstein & Rapp-Gabrielson, 1992). However, numerous gaps in our knowledge of molecular mechanisms of invading internal organs causing local and disseminated infection still remain at the present time (Zhou et al., 2016).

The conjugation process of creating markerless gene mutation is employed in an array of bacteria, which serves as a useful tool of constructing multi-gene scarless knock-outs. This facilitates in the construction of live attenuated vaccine strains and allows researchers to dissect the molecular mechanisms of virulence factors of H. parasuis and other bacterium (Heuermann & Haas, 1998; Oswald et al., 1999). A prerequisite for this process includes a marker-containing strain (Reyrat et al., 1998); a streptomycin resistance marker based on rpsL or rrs mutations is one of the most frequently used phenotypes (Kim et al., 2011).

Streptomycin (SM), belonging to aminoglycoside antibiotic, acts on the ribosome, inhibiting the translation of mRNA and therefore disrupting protein synthesis (Schatz, Bugie & Waksman, 2005). High-resolution melting and direct sequencing method in Mycobacterium tuberculosis and other species have been well established that streptomycin resistance mainly comes from mutations in the rpsL and rrs genes, which encode the ribosomal protein S12 and 16S rRNA, respectively (Wang et al., 2011). To generate SM resistant mutants by using spontaneous mutation, ultraviolet light (UV) induced mutations or transformation with plasmids containing streptomycin resistance markers has been used on various organisms (Timms et al., 1992; Skorupski & Taylor, 1996; Kim et al., 2011; Tsai et al., 2014). However, spontaneous or UV-induced mutants leading to a streptomycin resistance in H. parasuis probably occurs at a fairly low rate; SM resistant mutant generation in this organism was very inefficient in our hands using these strategies (data not shown). Moreover, the molecular mechanism of SM resistance is still unclear in this species.

EMS (ethyl methane sulfonate, methyl methanesulfonate) is a widely used chemical mutagen for inducing random mutations throughout the genome of plants (for example, Cucumis sativus) and some bacteria (for example, Escherichia coli) for its effective and efficient inductive effects (Parkhomchuk et al., 2009; Shah et al., 2015). Thus in this report, we introduced SM-resistance in H. parasuis based on the random somatic hypermutation effect of EMS and further performed site-directed point mutations in the same sites. We have previously demonstrated that arcA gene, encoding ArcA, a response regulator, contributes to the serum resistance and virulence in H. parasuis (Ding et al., 2016). ArcA and its sensor kinase ArcB, are one of the two-component signal transduction systems in H. parasuis. Since it is a global regulatory factor, whether it will affect natural transformation in H. parasuis has never been reported. tfox gene is a core competence gene-regulon (by regulating cAMP-CRP complex to positively act on CRP-S sites in the promoter regions of late-competence genes). It has been well established that bacteria would lost its natural transformability when tfox, whose homologue is sxy in H. influenzae, is deleted in H. influenzae and in many other bacteria (Sinha, Cameron & Redfield, 2009; Sinha, Mell & Redfield, 2012). However, the tfox gene cannot be deleted seamlessly by a previous technique established by us in that the successive markerless mutation system method requires a second round of natural transformation, whereas the △tfox mutant has lost its transformability at the end of the first round (Zhang et al., 2015).

In the light of these findings, we used site-directed mutations in rpsL gene that confer a complete streptomycin resistance as a positive selection marker to successfully create a double gene mutant (1401D88△tfox△arcA) in H. parasuis for further studying its biological functions. Moreover, we further used the genomic DNA of SM derivative to verify natural competence.

Material and Methods

Bacteria strains, plasmids and culture conditions

The bacteria strains, plasmids and primers used in this study are listed in Tables S1 and S2. Plasmids were propagated in E. coli DH5α or S17-1 (pir) and bacteria were grown in liquid Lysogeny Broth (LB, Difco, USA) medium or on LB agar (Invitrogen, Shanghai, China) plate. When required, the medium was supplemented with kanamycin (Kan; 50 µg ml−1) or ampicillin (Amp; 100 µg ml−1) from Sigma-Aldrich, USA. H. parasuis was routinely cultured in Tryptic Soy Broth (TSB; Difco, Franklin Lakes, NJ, USA) or on Tryptic Soy agar (TSA; Difco, Franklin Lakes, NJ, USA) supplemented with 5% inactivated bovine serum (Solarbio, Beijing, China) and 0.1% (w/v) nicotinamide adenine dinucleotide (NAD; Sigma-Aldrich, St. Louis, MO, USA) (TSB++ and TSA++). Where necessary, the media were supplemented with streptomycin (SM; 25 µg ml−1) or kanamycin (Kan; 50 µg ml−1). Unless otherwise stated, all strains were grown at 37 °C.

DNA manipulations

Bacterial genomic DNA was extracted using a TIANamp Bacteria DNA Kit (TIANGEN, Beijing, China). Small-scale plasmid DNA preparations were generated using a E.Z.N.A™ Plasmid Miniprep Kit (Omega Bio-Tek, Norcross, GA, USA). The DNA fragments were amplified in a C1000™ Thermal Cycler using the 2×Taq Plus Master Mix (Vazyme, Jiangsu Sheng, China) or I-5TM 2 ×High Fidelity Master Mix (MCLAB, South San Francisco, CA, USA). Purification of DNA fragments from the PCR reaction and the restriction digests were performed using the DNA Fragment Purification Kit Ver.4.0 (TaKaRa, Shiga, Japan). In-fusion segments were recombined to linearized pK18mobsacB (EcoRI/BamHI) using a ClonExpress II One Step Cloning Kit (Vazyme, Jiangsu Sheng, China). Restriction enzymes were purchased from TaKaRa. Point mutations were performed according to manual of Fast mutagenesis system Kit (TransGen Biotech, Beijing, China).

Antibiotic and Ethyl Methane Sulfonate (EMS) susceptibility assays

Assays on the minimal inhibitory concentration were carried out as described by Li (Li et al., 2016b). Briefly, overnight-grown bacteria were grown in TSB++ with twofold serially diluted SM (from 2 µg/mL to 128 µg/mL; MIC-S) in a 96-well microtiter plate. The plates were incubated at 37 °C for 24 h, after which, the results were scored for growth or no growth. The experiments were independently performed at least three times in triplicate. The MIC of EMS (from 0.5 mM to 40 mM; MIC-E) was analyzed in sterilized tubes. The results were also scored for growth or no growth. Moreover, the MIC-S of EMS induced SC1401 derivatives were also determined using the above method.

The minimal bactericidal concentrations of EMS for H. parasuis (MBC-E) were determined using the method modified from Li and Zhang (Zhang & Mah, 2008; Li et al., 2016b). Overnight-grown H. parasuis in sterilized tubes were exposed to different concentration of EMS (from 0.5 mM to 40 mM) for at least 12 h. Then, fresh TSB++ was used to replace the EMS medium and incubated for additional 16 h. The viability of bacteria was assessed by transferring a spot of the culture onto a TSA++ plate and incubation for 16 h. The experiments were independently performed at least three times in triplicate. Bacteria were serially passaged in TSB++ with MIC-E of EMS for five more generations for further determining the optimum inducing concentration of EMS.

In addition, the Oxford cup assay was used to evaluate the streptomycin resistance of wildtype H. parasuis SC1401, Hps32, Hps33 and derivatives 1401D88, 1401D43. Inocula were prepared and spread uniformly on TSA++ plates. One hundred microlitres of different concentrations (8,192 µg ml−1 and 4,096 µg ml−1 per well) of SM were added to the Oxford cups which were placed at equal distances above agar surfaces. The zone of inhibition for each concentration was measured after an extra 16 h-incubation at 37 °C. The experiments were independently performed at least three times in triplicate (Shang et al., 2014).

Ethyl Methane Sulfonate discontinuously inducing method

Wild type SC1401 (whose genomic sequence has been deposited in GenBank under the accession number NZ_CP015099.1), which was SM sensitive and used as the parental strain, was cultured in TSB++ with an optimum concentration (10 mM) of EMS (AR, Biotopped, China) at 37 °C with agitation for 12 h. The culture (F1) was transferred into 5 mL of fresh TSB++ without EMS, in a “Healing Resurgence” step. The F2 was subcultured and induced by 10 mM of EMS again with the same method for F1, F2, F3, F4 generations. Each generation was plated onto TSA++ with appropriate SM antibiotic and incubated for 24 h. The visible single colonies were propagated and the DNA extracted for further identification.

Sequence analysis

The 375-bp sequence of rpsL (locus tag: A4U84_RS04600) were amplified from the visible single colonies’ genomic DNA with primers rpsL-P1/P2 (Table S2). The PCR products were then purified. The products were cloned to pMD-19T(simple) for sequencing by BGI, Beijing, China. The RpsL protein sequences of H. parasuis were obtained by translating gene sequences using DNAMAN V6. Gene and protein sequences were then analyzed using BLAST in a comparison with that of wild type SC1401.

Site-directed mutagenesis

To further delineate mutations in rpsL, rather than other random mutations in whole genome, principally contributing to SM resistance induced. The entire rpsL coding sequence and corresponding flanking regions (975 bp) were amplified from genomic DNA of wild type SC1401 using primers rpsL-PM-P1/P2 and subsequently cloned into suicide vector pK18mobSacB, generating wild type pkrpsL. Afterwards, rpsL-43-P1/P2 and rpsL-88-P1/P2 were used to amplify the whole pkrpsL backbone with respective point mutations at codon 43rd (A128G) and 88th (A263G) in ORF of rpsL with 2 ×Transtart Fastpfu PCR SuperMix. Then the products were digested by DMT enzyme to move methylated template plasmids. Plasmids were subsequently mobilized into DMT competent cells by the CaCl2 method. The mutants were confirmed by sequencing.

The wild type pkrpsL (control), mutant plasmids pkrpsL43 (A128G) and pkrpsL88 (A263G) were transformed into wild type strain SC1401 by electroporation. The electroporation parameters were: 1800 v/mm, 25 µF, 200 Ω. The cells were screened by TSA++ containing 25 µg ml−1 of SM (although additional 10% sucrose together with antibiotic for screening transformants may be advantageous in order to prevent illegitimate recombinations, we found that once we obtained SM-resistant transformants, nearly all of them were expected derivatives after direct sequencing, even without the inclusion of 10% sucrose in the medium.). The visible single colonies were propagated and the DNA extracted and sequenced for further identification.

Construction of plasmids pkTLR and pkALR

Plasmids pkTLR and pkALR were constructed for tfox and arcA gene deletions and counter-selection in H. parasuis 1401D88, respectively. Here, we use pkTLR as an example to describe the process (Fig. 1). The 750-bp upstream homologous arm region and the 767-bp down homologous arm region of tfox were amplified using primers tfoxL-P1/P2 and tfoxR-P1/P2 from the genome of SC1401 (or 1401D88), respectively. These two flanking fragments were then purified using Qiaquick spin column (Qiagen, Hilden, Germany) and integrated by overlap extension PCR with tfoxL-P1/tfoxR-P2, after which the in-fusion segment was recombined to linearized pK18mobsacB (EcoRI/BamHI) using ClonExpress II. One Step Cloning Kit (Vazyme, Jiangsu Sheng, China). After confirmation by PCR and sequencing, the resulting plasmid, pkTLR, was mobilized into S17-1 (λpir) by the CaCl2 method.

Figure 1 Scheme for the construction of pkTLR.

Primer pairs tfoxL-P1/P2 and tfoxR-P1/P2 were used to amplify the upstream and downstream arm of tfox gene, respectively. Each with 21 bp homologous fragment for recombination to each other and to vector pK18mobSacB. Both products were integrated by overlap extension PCR with primers tfoxL-P1/tfoxR-P2. The resulting segment was then cloned into pK18mobSacB by a ClonExpress II One Step Cloning Kit (Vazyme, Jiangsu Sheng, China).

Construction of unmarked in-frame targeted mutant 1401D88ΔtfoxΔarcA of H. parasuis 1401D88

An unmarked tfox mutant was constructed using a two-step selective method (Fig. 2; Fig. 3). DNA was mobilized from E. coli S17-1 (λpir) to H. parasuis using a filter mating technique adapted by Oswald with some modification (Oswald et al., 1999). Briefly, S17-1 (λpir) harboring pkTLR and 1401D88 strains were grown to exponential phase in SOC and TSB++, respectively. The bacteria cultures were centrifuged at 4,350×g for 5 min; the pellet was resuspended in 1mL of TNM-buffer (1 mM Tris–HCl pH 7.2, 10 mM MgSO4, 100 mM NaCl). Aliquots corresponding to 0.5 ml of donor and 1 ml of recipient (each adjusted to an OD600 = 1) were mixed and plated onto a nitrocellulose filter membrane (0.45 µm pore size, 2.5 cm diameter; Millipore, Darmstadt, Germany). The NC membrane with bacterial mixture was placed onto TSA++ plates and incubated at 30 °C for 12 h. Eventually, the cultures were scraped up and spread on selective TSA++ supplemented with 25 µg ml−1 of SM and 50 µg ml−1 of Kan and incubated at 37 °C for 24 h. The transformants were then examined for sucrose sensitivity by plating on TSA++ supplemented with 10% sucrose (UP; Amresco, Solon, OH, USA) and 25 µg ml−1 of SM. PCR was carried out to confirm the appropriate deletion in colonies resistant to sucrose and sensitive to kanamycin. Afterwards, the mutants were continuously passed to 20 times on TSA++ and primers tfox-P1/P2, tfoxL-P1/tfoxR-P2 and Hps-P1/P2 were adopted for further confirming stability of the desired mutant.

Figure 2 Protocols of constructing in-frame mutants in H. parasuis using conjugation method.

Figure 3 Principles of obtaining gene mutant phenotype (in-frame mutation) by homologous recombination.

The whole pkTLR recombines to the tfox loci on the chromosome by the first allelic replacement thus the transformants can be selected by TSA++ supplemented with SM/Kan; through a second allelic replacement, we obtained mutants through PCR screening.

The effectiveness of our novel protocol for generating a double-gene mutant was confirmed when we further deleted the arcA gene from 1401D88△tfox. Likewise, the process was repeated as above, using a different primer set. The upstream and downstream homologous arms regions of arcA were amplified using primers arcAL-P1/P2 and arcAR-P1/P2, respectively. Primers arcA-P1/P2, arcAL-P1/arcAR-P2 and Hps-P1/P2 were adopted for further confirming the stability of the desired mutant.

The whole extract of 1401D88△tfox△arcA was further analyzed by western blotting. Briefly, the whole-cell lysates of the 1401D88, 1401D88△tfox△arcA mutant strain and SC1401 strain were analyzed by 12% SDS-PAGE electrotransferred to a polyvinylidene fluoride membrane (PVDF membrane). After being activated in methanol for 30 s and blocked with 5% BSA in TBST at room temperature (RT) for 30 min to 1 h, the membrane was incubated at RT for 1 h or overnight with mouse antiserum against recombinant Tfox or ArcA as the primary antibody, respectively. The membrane was rinsed 3–5 times with no less than 25 mL of TBST between incubations. Horseradish peroxidase (HRP)-conjugated rabbit-anti-mouse IgG (Earth, Hong Kong) were used as the secondary antibody (1:10,000 dil.). The membrane was finally developed with Immun-Star Western C Kit (Biorad, USA) according to the manufacturer’s instructions (Ding et al., 2016).

Using genomic DNA of 1401D88 and 1401D43 to verify natural transformation

The genomic DNAs of H. parasuis SC1401 derivatives 1401D88 and 1401D43 via site-directed point mutations in rpsL gene (as indicated in ‘DNA manipulations’) that confer SM resistance to H. parasuis were used to verify natural transformation capacity of H. parasuis strain SC1401; genomic DNA of SC1401△htrA, HPS32, HPS33 and the plasmid DNA pKBHK were used as controls. The protocol for natural transformation was performed as previously described with some modifications (Bigas et al., 2005). Briefly, recipient bacteria were grown in TSB++ to log phase or to an OD600 = 1.8 (about 2.8 ×109 cfu/mL). The bacteria were then spotted on TSA++, cultured overnight at 37 °C and resuspended in TSB++ at 20 ×1010 cfu/mL. A 20 µL aliquot of suspension was supplemented with 1 µg of donor DNA. The mixture was incubated for 10 min at 37 °C and then spread in a small area on TSA++. Cultures were incubated for an additional five hours at 37 °C to induce expression of antibiotic resistance. Afterwards, bacteria were harvested and resuspended and plated on TSA++ with SM in concentrations ranging from 25 µg ml-1 to no antibiotic. Bacteria were incubated for 36 h.

After incubation, the colonies on TSA++ and on selective plates were counted. Transformation frequencies were determined from the number of antibiotic-resistant CFU mL−1 divided by the total CFU mL−1 scored on nonselective agar. Transformation efficiency was evaluated by calculating number of transformed CFU per µg of donor DNA.

Statistical analysis

The statistical analysis was performed using R studio (loaded with R V3.3.2) for Windows (RStudio Team, 2015). Comparison of several test series was evaluated by analysis of variance (ANOVA). The significance of differences between groups was calculated using Student’s t-test. A P value < 0.05 was considered to be statistically significant (*), and <0.01 highly significant (**).

Results

Determination of MIC & MBC of streptomycin and ethyl methane sulfonate in H. parasuis

To investigate an appropriate inhibiting concentration of SM and EMS, we determined minimal inhibitory concentration of SM (MIC-S) and EMS (MIC-E) for H. parasuis SC1401. The results showed that MIC-S for SC1401 to be 16 µg ml−1; however, 24 µg ml−1 of SM was bactericidal (Table 1). Hence 25 µg ml−1 of SM was employed in later experiments to screen out SM-resistant derivatives from wild type SC1401. By comparison, EMS was shown to have fairly strong antibacterial activity. As shown in Fig. 4, we could see flocculent bacteria when the culture was exposed to 10 mM of EMS, but not in the TSB++ with 20 mM of EMS. Therefore, twenty millimoles per liter of EMS was determined to be the MIC-E of H. parasuis SC1401 (Table 1 and Fig. 4).

Table 1 Susceptibility of H. parasuis strains to streptomycin (SM) and ethyl methane sulfonate (EMS).

Object	Test indexa	Strainsb	
		SC1401	1401D43	1401D88	HPS32	HPS33	
SM	MIC-S (µg/mL)	16	>4,096	>4,096	16	16	
MBC-S (µg/mL)	24	>4,096	>4,096	28	32	
EMS	MIC-E (mM)	20	–	–	–	–	
MBC-E (mM)	30	–	–	–	–	
Notes.

a MIC-S/E, minimal inhibitory concentration of SM/EMS; MBC-S/E, minimal bactericidal concentrations of SM/EMS.

b 1401D43 and 1401D88 are completely resistant to SM.

Figure 4 High concentration of EMS exhibits obvious bacteriostatic activity.

Visible flocculent bacteria under 10 millimoles per liter of EMS were more easily subjected to discontinuous induction; these were serially passaged.

To determine optimal inducing, subinhibitory concentrations of EMS, we tested the minimal bactericidal concentrations (MBC-E). Overnight-grown seed broth was subcultured in fresh TSB++ with different concentration of EMS (from 0.5 mM to 40 mM) and subcultured again with TSB++ with no EMS. The results showed that 30 mM of EMS was the definitive MBC-E (Table 1).

The EMS-induced derivatives of SC1401 and MIC-S assay

As shown in Table 1, thirty mili-moles per liter of EMS has strong bactericidal activity (this concentration was determined to be the MBC-E), whereas 10 mM of EMS was shown to be the MIC-E (Fig. 4 and Fig. S1). Therefore, in consideration of maximizing the inductive effect of EMS, we used a discontinuous induction method in our experiments. The culture in a 10 mM of EMS-containing TSB++ was transferred into 5 mL of fresh TSB++ without any EMS, in a “Healing Resurgence” step of our protocol. Each generation was plated onto TSA++ with appropriate SM antibiotic and incubated for 24 h. In this way, we obtained SM-resistant generations in F7, F9 and F11. Among them, mutations in rpsL at codon 43rd (A128G) and 88th (A263G) were confirmed by sequencing.

By comparison, these two kinds of mutations conferred complete resistance to SM (Table 1). The two derivatives can survive well at a concentration of at least 4,096 µg ml−1 of SM. As a control, H. parasuis wild type strains SC1401, HPS32, HPS33 can only survive in less than 16 µg ml−1 of SM/TSB++. We also performed the Oxford cup assay to visually demonstrate different levels of SM-resistance. As shown in Fig. 5, 1401D43 and 1401D88 can form complete bacteria lawns on TSA++ with both 100 µl of 8,192 µg ml−1 and 4,096 µg ml−1 of SM per well, while inhibition zones can be easily recognized for wild type H. parasuis, indicating the successfully artificially EMS-induced products were far more resistant to SM than wild type SC1401.

Figure 5 Streptomycin resistance analysis based on Oxford cup assay.

Oxford cups containing 8,192 µg ml−1 and 4,096 µg ml−1 per well of SM were used to analysis bacterial resistance. It demonstrated that (D, E) the derivatives 1401D43 and 1401D88 were completely resistant to SM, whereas (A–C) wild type H. parasuis exhibited distinct bacterial inhibition zones.

rpsL hot mutations

It has been well established in procaryotic organisms that streptomycin hinders the functioning of ribosome by binding to 16S rRNA helices and ribosomal protein S12, encoded by rrs and rpsL, respectively. SM also interacts with decoding site and shifts 16S rRNA helix in the direction of ribosomal protein S12 which in turn induces miscoding by stabilising the closed confirmation of 30S ribosomal subunit (Suriyanarayanan et al., 2016). We directly sequenced the rpsL and rrs genes for wild-type strains Hps32, Hps32 and artificially induced SM-resistant strains. We used BLAST and these reads to query the rpsL and rrs sequences of H. parasuis SC1401. The results of DNA sequencing showed that all these SM-resistant strains harbored rpsL mutations. Comparatively, point mutations at codon 88th (AAA → AGA; K88R) and at codon 43rd (AAA → AGA; K43R) confer much higher SM-resistance on the EMS-induced derivatives than H. parasuis wild type strains Hps32, Hps32, in which random mutations of synonymous codon have been introduced (Fig. 6), indicating that the mutations at codon 43rd and 88th are the core regions blocking SM binding in H. parasuis, whilst other point mutations occurred less frequently. The mostly highly-resistant SM mutations occur in these two codons. However, we haven’t found a derivative with simultaneous mutations in these two sites. By comparison, rrs mutation could be found in derivative 1401D43 (in which only G920A have been identified), HPS32 and HPS33 (in which almost fifteen point-mutations have been identified, data not shown), but obviously not a greater factor of inducing the high resistance to SM (Table 1 and Fig. 5). We didn’t find any mutations in rrs in 1401D88.

Figure 6 Mutation sites in rpsL gene of derivatives 1401D43 and 1401D88 as well as two wild type SM-resistant H. parasuis.

The hot mutation sites in H. parasuis indicates that the K→R mutations in 43rd and 88th codons in RpsL conferred resistance to SM. While we could only find synonymous mutations in wild type HPS32 and HPS33 within rpsL gene, so their SM-resistance may come from the rrs mutations (have been identified but not shown) or other genetic factors. However, these kinds of mutations or factors contribute little in conferring SM-resistance.

Identification of successful construction of pkTLR and pkALR

Further investigation of Kan-resistant colonies showed that the in-fusion segments of up/downstream regions of tfox was successfully integrated into linearized pK18mobsacB. PCR runs with primers tfoxL-P1/tfoxL-P2, tfoxR-P1/tfoxR-P2 and tfoxL-P1/tfoxR-P2 demonstrated that the products were amplified as expected, which was followed by sequencing for further confirmation (Fig. S2). The plasmid pkTLR was then used to create unmarked in-frame tfox mutations in H. parasuis. Likewise, the plasmid pkALR was generated and identified as above. PCR runs were carried out using primers arcAL-P1/arcAL-P2, arcAR-P1/arcAR-P2 and arcAL-P1/arcAR-P2.

Construction of an unmarked mutation of tfox and arcA genes in H. parasuis 1401D88

Based on a two-step natural transformation mutagenesis method, we used a ‘dual’ process with one round of SM selection and a subsequent sucrose counterselection. The in-fusion segment was successfully transferred to the chromosomal DNA of recipient strain and subsequently integrated into the host tfox loci through homologous recombination, yielding the in situ deletion. Here, the method of EMS induced high level-resistance to SM plays an important role in weeding out donor strain S17-1 (λpir), thus allowing us to generate unmarked mutants in H. parasuis that are multi-drug susceptible.

The plasmid pkTLR was integrated into the two flanking regions of tfox gene at the genomic site, so that the original tfox gene was replaced seamlessly. SM and Kan positive selection was highly successful, and the PCR tests validated both the insertion of the kan cassette and SacB gene in most of the Kan-resistant clones (Fig. S3A). In the second round of homologous recombination, the backbone of pk18mobSacB was removed, followed by sucrose counterselection and kanamycin susceptibility testing. PCRs were carried out to verify the appropriate deletion in colonies resistant to 10% sucrose and 25 µg ml−1 of SM but sensitive to Kan (Fig. S3B).

The second round is to remove arcA gene from 1401D88△tfox. Likewise, pkALR was used to create in-frame arcA knock-out mutant. PCR runs were carried out to confirm the appropriate deletion in transformants resistant to sucrose and 25 µg ml−1 of SM but sensitive to 20 µg/mL Kan.

Confirmation of 1401D88△tfox△arcA by western blotting

Western blotting confirmed that tfoX and arcA can be detected in the whole-cell extract of wild type H. parasuis SC1401 and 1401D88, but not in 1401D88△tfox△arcA (Fig. 7). This result further confirms the successful deletion of tfox and arcA genes in strain 1401D88.

Figure 7 Western blotting analysis of wild type H. parasuis and mutant.

The whole-cell lysates of the 1401D88, mutant and SC1401 strains were detected using tfoX and arcA mouse antiserum, respectively. The derivatives 1401D88 displayed a similar band to the parent strain SC1401, while 1401D88△tfox△arcA strain did not. (A) Western blotting analysis using tfoX mouse antiserum as primary antibody. Lane 1, 1401D88 strain; Lane 2, 1401D88△tfox△arcA strain; Lane 3, SC1401 strain. (B) Western blotting analysis using arcA mouse antiserum as primary antibody. Lane 1, 1401D88 strain; Lane 2, 1401D88△tfox△arcA strain; Lane 3, SC1401 strain.

Verification of natural competence

Point mutations in rpsL which don’t confer negative effect on growth were assumed to be more effective in verifying natural competence (see Fig. S4 for comparison of growth rates of H. parasuis mutants and wild type SC1401). To confirm this hypothesis, we tested the natural frequencies of different donor DNA, including genomic DNA of H. parasuis 1401D43, 1401D88, SC1401△htrA (in which the Kan cassette had been stably inserted.) and the plasmid DNA pKBHK which was previously used to creat htrA knock-out mutant in H. parasuis and to screen natural competent cells, as well as the naturally SM-resistant strains HPS32 and HPS33. The results showed that transformation frequencies of 1401D43, 1401D88, SC1401△htrA genomic DNA after an extended growth period (13 h) on TSA++ were considerably higher than those for pKBHK construct. Genomic DNA of HPS32 and HPS32 couldn’t confer SM-resistance to SC1401, which also in support of the elucidation highlighted in the “Introduction” that spontaneous mutations in this species occur at a fairly low level. The transformation frequencies of derivative genomic DNA reached a fairly high level (about 2.458 × 10−3) compared to SC1401△htrA genomic DNA. However, the highest transformation frequency of genomic DNA of SC1401△htrA reached to 4.748 × 10−4 (Fig. 8). Furthermore, genomic DNA of 1401D43, 1401D88 can be actively taken up by recipient cells. The transformation frequencies of 1401D43, 1401D88 DNA derived from point mutations in SC1401 are significantly higher than SC1401△htrA genomic DNA which carries a KanR cassette in the htrA loci. In independent repeated experiments, more than thirty transformants were directly sequenced for further identification of desired mutations in rpsL.

Figure 8 Natural transformation frequency of different donor DNA.

Lines are achieved by mean values of three replicates fitting, and shaded areas indicate the confidence interval between three replicates.

Discussion

H. parasuis is one of the most economically significant swine pathogens at present in the swine industry throughout the world. This pathogen usually invades barrier of blood vessels and causes multiple syndromes when the host’s immunity is suppressed. At least 40 H. parasuis species have been sequenced and large array of potential virulence factors have been found (Li et al., 2016a); however, their pathogenic mechanisms and evolution remain largely unknown. Previous studies on gene deletion usually based on natural transformation system of highly transformable strains, such as SC1401 (Zhang et al., 2016), SC096 (Zhang et al., 2014a), EP3 (Ding et al., 2016), CF7066 (Huang et al., 2016). However, the natural transformation technology relies heavily on screening of natural competent cells, which is inefficient since not all bacteria in H. parasuis are competent cells, at least under the existent experiment cues and conditions (Seitz & Blokesch, 2013; Johnston et al., 2014).

Although EMS may introduce hypermutations in H. parasuis, certain concentration of EMS (more than 30 mM) exhibits a bactericidal effect, thus we tested the MIC-E and MBC-E and subsequently found 10 mM of EMS is an optimized concentration for inducing rpsL mutations H. parasuis. We couldn’t obtain SM-resistant derivatives in the earlier attempts using EMS. A discontinuous induction method in which one alternate step labelled “Healing Resurgence” was applied. With this technique, we successfully obtained an array of SM-resistant derivatives, in which K43R and K88R were thought to be the most important mutations blocking SM binding sites, as described in Mycobacterium tuberculosis and other species (Arjomandzadegan & Gravand, 2015). Followed by site-directed mutagenesis system, we successfully obtained two derivatives 1401D43 and 1401D88 which are completely resistant to SM from two clinical H. parasuis isolates and wild type SC1401. These mutants facilitate our later conjugation process of creating unmarked tfox and arcA knock-out mutants in 1401D88. The hot mutation sites in H. parasuis also indicates that the K → R mutations in 43rd and 88th codons in rpsL, which encodes the highly conserved RpsL protein of the ribosomal accuracy centre (Toivonen, Boocock & Jacobs, 1999), have a greater propensity to confer drug resistance to streptomycin than rrs and other mutation sites. Since we failed to obtain a derivative with simultaneous mutations in both sites, we postulated that mutation at one codon is enough to confer complete SM resistance (Table 1), with no need for generating both point mutations in a same strain, or that mutations at both sites simultaneously are not efficient. It is noteworthy that the unmarked mutation system in this report is not a strictly markerless mutation method from the point of the whole genome, since we have introduced a SM resistance in this strain through the mutation in rpsL gene.

We used genomic DNA of 1401D43, 1401D88, and SC1401△htrA, as well as plasmid DNA pKBHK to transform wild type SC1401 to test the transformation frequencies. The results demonstrate that the transformation frequencies of genomic DNA of 1401D43 and 1401D88 occur at a high level (about 2.458 ×10−3), nearly four times higher than that of SC1401△htrA (about 4.748 ×10−4). We postulated that point mutations that don’t generate negative effects on growth will presumably to integrate into host’s genome more effficiently under the RecA-driven homologous recombination. Since the genomic DNA of 1401D43 and 1401D88 can be actively and efficiently taken up by natural competent cells, DNA with antibiotic-resistant markers could be used to screen out natural competent bacterium or to verify natural transformation in HGT, since rpsL gene is highly conserved in H. parasuis, but not htrA gene, which is a differentially expressed gene (DEGs) in strains isolated from the lung of a diseased pig and an avirulent strain isolated from the nasal swab of a healthy pig (Zhang et al., 2014b).

As a whole, a modification of an EMS (ethyl methane sulfonate) system was applied to generate streptomycin (SM) resistant in H. parasuis based on its efficient mutagenesis which may exhibit stochasticity in the molecular processes at the single-cell level (Parkhomchuk et al., 2009). DNA sequencing was performed to identify a mutation occurring in rpsL at codons 43rd and 88th that confer SM-resistance. Site-direct mutagenesis was applied to generate the same point mutation in clinical H. parasuis isolates SC1401. The derived H. parasuis 1401D88 bearing single point mutation in rpsL gene was used to generate a double mutation (△tfox△arcA). Specially, competence-specific tfox gene (the orthologue of sxy in H. influenzae) was reported to be one of the most important regulons controlling natural competence in H. influenzae (Cameron et al., 2008), Aggregatibacter (formerly known as Actinobacillus) actinomycetemcomitans (Bhattacharjee, Fine & Figurski, 2007) and V. cholerae (Meibom et al., 2005; Johnston et al., 2014). Another gene, arcA, encoding ArcA, a response regulator, and a sensor kinase ArcB, is one of the three two-component signal transduction systems (TCSTS) found in H. parasuis. We have previously demonstrated that arcA gene contributes to the serum resistance and virulence in this species in a previous study (Ding et al., 2016). Furthermore, the genomic DNA of the highly resistant derivatives 1401D43 and 1401D88 could be used as a useful tool to verify natural transformation, thus enabling us to verify naturally competent cells or study horizonal gene transfer (HGT).

Conclusions

In summary, our new novel development of EMS-induced SM-resistant method and site-directed point mutations in H. parasuis provided a new avenue for genetic manipulation in this bacterium. We have established a discontinuous induction method of EMS for triggering somatic hypermutation in random sites in H. parasuis, and successfully screened out hot point mutations in rpsL which confer complete SM resistance upon transformants. The DNA containing the desired mutations could be used as a positive selection marker to create an unmarked in-frame tfox and arcA double knock-out mutants which cannot be established through natural transformation. Finally, we confirmed that the genomic DNA harboring rpsL hot mutations was effective and efficient in verifying natural transformation, which will facilitate the identification of natural competent cells and further studies on the mechanism of natural competence in H. parasuis.

Supplemental Information

Data S1 Raw data

Click here for additional data file.

Supplemental Information 1 Supplementary files

Click here for additional data file.

Additional Information and Declarations

Competing Interests

Author Contributions

Data Availability

The authors declare there are no competing interests.

Ke Dai conceived and designed the experiments, performed the experiments, analyzed the data, wrote the paper.

Xintian Wen conceived and designed the experiments, wrote the paper.

Yung-Fu Chang conceived and designed the experiments, analyzed the data, wrote the paper, reviewed drafts of the paper.

Sanjie Cao, Qin Zhao and Xiaobo Huang performed the experiments.

Rui Wu, Yong Huang, Qigui Yan and Xinfeng Han analyzed the data.

Xiaoping Ma prepared figures and/or tables.

Yiping Wen prepared figures and/or tables, reviewed drafts of the paper.

The following information was supplied regarding data availability:

The raw data has been provided as a Supplemental File.

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
