# Peer review of "A streptomycin resistance marker in H. parasuis based on site-directed mutations in rpsL gene to perform unmarked in-frame mutations and to verify natural transformation"

_PeerJ, doi:10.7717/peerj.4253_

## Round 0.1 · original submission · Major Revisions

Please address all comments, especially those related to the description of the methods (which are presently quite confusing, lack an explanatory figure and are not clear on whether a positive selection or a counter-selectable marker was used), the conflation of results/methods in the introduction section (lines 65-85) and the (unremarkable, according to reviewer #3) bactericidal/bacteriostatic effect of EMS at different concentrations (lines 357-359), which should be expected from its teratogenic action.

You mention that your method is promising and that "Combined with natural transformation-based knockout system and this genetic technique, multiple deletion mutants or attenuated strains of H. parasuis can be easily constructed." I cannot help but think that your manuscript would benefit enormously by the inclusion of one such example, as additional proof of principle. My final decision will not depend on such an inclusion, which is likely to be laborious (for the demonstration of phenotypic effects, if not for the construction step) but would, in my opinion, increase the attractiveness of your method to readers and reviewers.


Some authors reply to reviewers in a table format, which I do not find easy to read. When you prepare your rebuttal, please provide the full text of all of the reviewers' comments to the initial version of this submission, interspersed with your detailed replies to each point (preferably in a different font, for ease of reading) instead.

PeerJ request re-submissions to be accompanied by a copy of the manuscript file with highlighted changes. Please do not highlight those changes manually: use your word-processor built-in "track changes feature" instead, to compare the initial submission to your modified manuscript.

Reviewer 1 ·

Basic reporting

The paper is not organized well. For example, there is no connection between section "3.4" and "3.3".

Experimental design

The experimental design is strange. I don't know why the author using EMS to induce mutation. In addition, the author did not tell us how many mutant they have gotten and what are they.

Validity of the findings

The authors claimed that they have developed a new novel development of EMS-induced SM-resistant method. But I do not think so.

Additional comments

Some experiment details should be described in the methods section instead of in the results section.

Reviewer 2 ·

Basic reporting

1 Writing of a scientific article should be concise and clear. However, the logic of this paper is vague. Numerous grammatical errors also hinder understanding of the paper.
2 Lines 50-52, “A prerequisite for this process includes a marker-containing strain (Reyrat et al., 1998); a streptomycin resistance marker based on rpsL or rrs mutations is one of the most frequently used phenotypes (Tsai et al., 2014).”
The authors mixed up the concepts of positive selection marker and counterselectable marker. Streptomycin sensitivity is one of the most-used counterselectable markers in allelic exchange as described in the papers that the authors cited. To this end, a streptomycin-resistant strain and a suicide plasmid containing a wild-type rpsL gene are required. When the suicide plasmid is used to deliver an inactivated allele of the target gene in the chromosome, expression of the wild-type rpsL gene on the integrated plasmid confers a streptomycin-sensitive phenotype upon streptomycin-resistant strains and allows for selection with streptomycin to detect loss of the vector. However, in this study, the authors used sucrose sensitivity, which is conferred by the sacB gene on the suicide plasmid pK18mobSacB, instead of streptomycin sensitivity in the selection of the deletion mutant (lines184-201). Streptomycin-resistance only served as a positive selection marker for the wild type strain. In fact, any antibiotic to which the SC1401 strain is resistant but E. coli S17-1 is sensitive can serve as a positive selection. Did the authors analyze and compare drug resistance profiles of both strains before starting the study?

Experimental design

1 Lines 168-170, “ The cells were screened by TSA++ containing 25 μg ml-1 of SM. The visible single colonies were propagated and the DNA extracted and sequenced for further identification.”
Did the authors identify the mutants with genomic DNA sequencing?!! The corresponding data were not presented in the section of Result. In this experiment, streptomycin selection is not enough for allelic exchange because the frequency of double crossover events may be low and illegitimate recombination may occur. Addition of 10% sucrose to the selection medium is appropriate.

2 Lines 221-222, “The visible single colonies were identified by PCR and western blotting.” Why and how did the authors identify the colonies with western blotting assay?

Validity of the findings

1 Lines 320-323, please describe the assay of Western blot in the Section of Material and Methods.
2 Lines 326-327, “Point mutations in rpsL which don't confer negative effect on growth were assumed to be more effective in verifying natural competence.” Did the authors compare the growth rates of the wild type and the mutants?
3 Lines 334-336, “Genomic DNA of HPS32 and HPS32 couldn’t confer SM-resistance to SC1401, which also in support of our elucidation highlighted in “Introduction” that spontaneous mutations in this species occur at a fairly low level.” Why? Is transformation the cause of spontaneous mutation?

Reviewer 3 ·

Basic reporting

The level of English is acceptable. I found a few spelling errors and non-idiomatic expressions.

The Introduction and background sections will have to be improved. The references from line 41,42 below
should be substituted for the relevant original sources.

40 producing significant mortality and morbidity in pig farms and leading to serious economic losses
41 in the pork industry throughout the world (Yue et al., 2009;Zhang et al., 2016b).

Structure conforms to PeerJ standards as far as I can judge. Abstract is within the specified limits.

The manuscript has six figures which seem a bit excessive as I think the manuscript lacks an important figure.
This figure should be a scheme that shows how the so called "in-frame" deletions happen. This figure should
contain relevant plasmids and primers. I think figure 2 could be put among the suplemental figures to make room for this new figure.

All relevant raw data was supplied.

Experimental design is adequate and the findings are valid.

# Specific line by line comments


27 Combined with natural transformation-based knockout system and this
28 genetic technique, multiple deletion mutants or attenuated strains of H. parasuis can be easily
29 constructed

correct to:

Combined with a natural transformation-based...
* * *
38 pleomorphic, NAD-dependent opportunistic bacterium

Please explain what a NAD dependent bacteria is? Is it auxotrophic?
* * *
58 2011). To generate SM mutants by using

correct to:

58 2011). To generate SM resistant mutants by using
* * *
61 et al., 2014). However, spontaneous or UV-induced mutants of streptomycin resistance in H.
62 parasuis probably occurs at a fairly low rate in H. parasuis; SM mutant generation in this organism
63 is very inefficient using these strategies (data not shown).

correct to:

61 et al., 2014). However, spontaneous or UV-induced mutations leading to streptomycin resistance in H.
62 parasuis probably occurs at a fairly low rate; SM mutant generation in this organism
63 was very inefficient in our hands using these strategies (data not shown).


The statement is too strong give the lack of experimental detail.
* * *
The lines 65 - 85 do not belong in the materials section. Some should go to introduction and some to results.
* * *
90 S2. Plasmids were propagated in E. coli DH5α or S17-1 (λpir) and grown in liquid Luria-Bertani

LB means Lysogeny Broth and NOT Luria-Bertani.
* * *
118 growth. Moreover, the MIC-S of EMS induced SC1401 derivatives were also determined using
119 above method.

correct to:


118 growth. Moreover, the MIC-S of EMS induced SC1401 derivatives were also determined using
119 the above method.
* * *
138 Wild type SC1401 (deposited in GenBank under the accession NO. NZ_CP015099.1), which

The strain was not deposited, what was depositied was the genomic sequence.
* * *
The sections
2.7. Construction of Plasmids pkTLR and pkALR and
2.8. Construction of Unmarked In-Frame Targeted Mutant 1401D88ΔtfoxΔarcA of H. parasuis

Would benefit from the extra figure that I mentioned previously. It is very hard to follow the details of the genetic designs.
* * *
210 The genomic DNAs of H. parasuis SC1401 derivatives 1401D88 and 1401D43 via site-
211 directed point mutation were used to verify natural transformation capacity of H. parasuis strain

This is a mix of concepts that is hard to understand. What was done vis site-directed mutagenesis?
Rephrase to clarify.
* * *
255 we used a discontinuous induction method in our experiments. The culture in a 10mM of EMS-

Please correct spaces between dimension and units here and possibly throughout the manuscript.
* * *
The section between lines 271 - 289 needs to be split up to introduction and results or discussion
* * *
357 In this study, we found that EMS has a characteristic dual function in H. parasuis. High
358 concentration of EMS inhibits growth (20mM) or even exhibits a bactericidal effect (30mM),
359 whereas low concentration introduces random somatic hypermutations in genome.

Although, this is not directly my field this must surely be something that is observed every time
EMS is used. I remember performing experiments as an undergraduate to demonstrate that the a cmpound
would be bacteriostatic at low concentration and bactericidal at higher. Please consider removing this.
* * *
Experimental design

Experimental design is adequate. See above for more details.

Validity of the findings

Findings are valid. See above for more details.

---

## Round 0.2 · Minor Revisions

Your manuscript has been much improved. Please address the final comments by Reviewer #2

Reviewer 2 ·

Basic reporting

The manuscript still needs a thorough proofreading to improve the English language used.

Experimental design

In my comment that “In this experiment, streptomycin selection is not enough for allelic exchange because the frequency of double crossover events may be low and illegitimate recombination may occur. Addition of 10% sucrose to the selection medium is appropriate.” I mean the experiment of “2.6. Site-Directed Mutagenesis”, not the experiment of “2.8. Construction of Unmarked In-Frame Targeted Mutant 1401D88ΔtfoxΔarcA of H. parasuis 1401D88”. In 2.8, 10% sucrose was added into the selection medium. However, In 2.6, the sucrose was not used in the selection medium. Why?

Validity of the findings

No comment

---

## Round 0.3 · Minor Revisions

I have two requests for changes in your recent resubmission. :
- I would like you to include ( in the "response to referee" letter ) the response to the point I raised regarding the mention in the introduction of why the tfox and arcA genes were selected for the double mutant. I think your language is often much more clear in responses to reviewers (as in the accompanying response to referees) than in the text that ends up in the manuscript, probably because you feel more comfortable in the relatively informal language register of a back-and-forth conversation than in the more formal style of the manuscript. There is a general tendency (among authors) to try to explain things as concisely as possible, but in some cases, like this one, extreme concision gets in the way of conveying the message. As a rule, I think authors should not be afraid to provide longer explanations in the manuscript. After all, if some of those explanations prove to be unnecessary, some reviewer or editor will surely point that out, but if the text is too short reviewers may simply give up on understanding and develop a prejudice against the paper.

- The language you used in your response to the point raised by reviewer #2 is confusing as it can be read to mean two different things:
1st interpretation: no sucrose was included , but sequencing showed that the mutants produced were the ones intended. Adding sucrose would be advantageous and recommendation to include 10% sucrose in this step was included to facilitate future reproductio ofthe experiment.

2nd interpretation: 10% sucrose was included although additional trials showed it to be unnecessary, since sequencing showed that the mutants produced were the ones intended even without sucrose.


Replace current text by the precise description of your methods and reasons why you can confidently claim that the correct mutants are generated in suffiicient abundance in the absence of sucrose (as explained to me in your previous email). You may suggest (like the reviewer) that 10% sucrose be addded to the medium in that step to ensure higher yields, but clarify whther or not you indeed used it.

You moved the complete text of lines 65-85 (of the previous version) to the conclusions. As a consequence, no explanation of the relevance of the tfoX and arcA genes now appears either in the "introduction" or the "materials" section, and this makes the reading of your manuscript more puzzling than before. Please introduce the relevance of those genes in the Introduction, and highlight it when discussing the selection of the genes to be deleted

---

## Round 0.4 · Minor Revisions

I am quite satisfied with your responses to the reviewers. There are still a few language issues which should be addressed before acceptance:

in line 58, replace "We have demonstrated " with "We have previously demonstrated "

in line 61, replace "Since it’s a global regulatory factor, whether it’ ll" with "Since it is a global regulatory factor, whether it will

in line 67, include the appropriate reference after "by a previous technique established by us "

in line 152 , replace the text "(although point mutations are easy to be obtained in our hands, but additional 10% sucrose together with antibiotic for screening transformants would be more advantageous in order to prevent illegitimate recombinations, even if there is a small probability.)" by a reworking of your response to reviewers "Although additional 10% sucrose together with antibiotic for screening transformants may be advantageous in order to prevent illegitimate recombinations, we found that once we obtained SM-resistant transformants, nearly all of them were expected derivatives after direct sequencing, even without the inclusion of 10% sucrose in the medium."

---

## Round 0.5 · accepted · Accept

Final revisions were performed as requested. I am glad to accept your paper for publication in PeerJ.